# SIMILARITY AND GENERALIZATION: FROM NOISE TO CORRUPTION

## ABSTRACT

Contrastive learning aims to extract distinctive features from data by finding an embedding representation where similar samples are close to each other and different ones are far apart. We study how neural networks generalize in similarity learning in the presence of noise, investigating two phenomena: Double Descent (DD) behavior and online/offline correspondence. We focus on the simplest contrastive learning representative: Siamese Neural Networks (SNNs). We introduce two representative noise sources that can act on SNNs: Pair Label Noise (PLN) and Single Label Noise (SLN). The effect of SLN is asymmetric, but it preserves similarity relations, while PLN is symmetric but breaks transitivity. We find that DD also appears in SNNs and is exacerbated by noise. We show that the density of pairs in the dataset crucially affects generalization. Training SNNs on sparse datasets affected by the same amount of PLN or SLN gives the same performance. On the contrary, using dense datasets, PLN cases generalize worse than SLN ones in the overparametrized region. Indeed, in this regime, PLN similarity violation becomes macroscopical, corrupting the dataset to the point where complete overfitting cannot be achieved. We call this phenomenon *Density-Induced Break of Similarity (DIBS)*. Probing the equivalence between online optimization and offline generalization in SNNs, we find that their correspondence breaks down in the presence of label noise for all the scenarios considered.

## 1 INTRODUCTION

In recent years, several works have studied generalization in neural networks (NNs) and the connection between the classical underparametrized regime, where the number of training samples is larger than the number of parameters in the model, and that of deep learning, where the opposite is usually the norm. Indeed, the empirical success of overparameterized NNs challenges conventional wisdom in classical statistical learning as it is widely known among practitioners that larger models (with more parameters) often obtain better generalization: Szegedy et al. (2015); Huang et al. (2019); Radford et al. (2019).

Two frameworks adopted to study generalization in regression or classification tasks are *Double Descent* (DD) and *online/offline learning correspondence*, which we describe in the following. DD from Belkin et al. (2019) connects "classical" and "modern" machine learning by observing that once the model complexity is large enough to interpolate the dataset (i.e., when the training error reaches zero), the test error decreases again, reducing the generalization gap. This pattern has been empirically observed for several models and datasets, ranging from linear models, as in Loog et al. (2020), to modern DNNs, as in Spigler et al. (2019); Nakkiran et al. (2020a). Instead, the online/offline learning correspondence of Nakkiran et al. (2021), studies the relationship between online optimization and offline generalization. The conjecture, empirically verified on supervised image classification, states that generalization in an offline setting can be effectively reduced to an optimization problem in the infinite-data limit. This means that online and offline test errors coincide if the NN is trained for a fixed number of weight updates. This setup aims to find a *connection* between under- and overparameterized models: the infinite-data limit (online) sits in the under-parameterized region (number of samples > number of parameters), while the finite-data case (offline) corresponds to the overparameterized regime (number of samples < number of parameters). Here, we test if this correspondence is also valid for similarity tasks. *DD and online/offline correspondence are two complementary approaches that look at different generalization properties:* while DD studies how the

network adjusts to an increasing number of parameters, online/offline training compares the network performance by varying the dataset size while fixing the number of weight updates. Although these approaches have mainly been applied to classification and regression, if they are associated with some fundamental properties of DNNs, they should also hold for other tasks such as similarity learning.

There are key differences between similar-different discrimination and classification. For similarity learning, the relation among features is crucial but not necessarily the features themselves. For this reason, a priori it is not possible to predict whether the DD behavior and the online/offline learning correspondence will also occur for similarity problems. To take the first steps towards understanding how DNNs generalize in similarity learning, we export both frameworks to the simplest contrastive learning representative, Siamese Neural Networks (SNNs) from Bromley et al. (1994); Chopra et al. (2005). A Siamese architecture is made of two identical networks sharing weights and biases that are simultaneously updated during supervised training. The two networks are connected by a final layer, which computes the distance between branch outputs. SNNs are trained using pairs of data that are labeled as similar or different. The task of a successfully trained network is to decide if the pair samples belong to the same class. Studying the DD and online/offline correspondence in SNNs and comparing the results with those found in classification problems requires identifying which properties/characteristics of the training set most influence similarity learning. We identified two crucial sources of variability: *(i)* the effect of noisy data in SNNs, and *(ii)* the density of pairs in the training set.

Noise is crucial in understanding generalization as it appears in every real-world dataset and may compromise model performance. While DD was also studied in the presence of noise,[1] very little (if none) attention was devoted to noise in the online/offline setting. By construction, SNNs can be affected by more complex types of noise than classification problems. This derives from the use of pairwise relations defining a similarity graph. To show the reaction of SNNs to different noise sources, we introduce two representative examples with distinctive properties: Single Label Noise (SLN) and Pair Label Noise (PLN), which we extensively describe in Sec. 2 and illustrate in the top panel of Fig. 1. As we will show, SLN breaks similar/different pairs balancing but preserves similarity relations. Instead, PLN acts symmetrically on pair labels, but it breaks transitivity and, thus, similarity. Furthermore, we show that similarity learning is strongly influenced by the *density of pairs in the training set*. In particular, we will show how pairs created from populations with different levels of similarity graph density/image diversity, i.e., the average number of different images appearing in a set of pairs, give rise to very different learning models. We discuss sparse and dense connections in detail in Sec. 2.

Our results show that

- DD clearly appears in SNNs, regardless of the noise level, a phenomenon rarely found in classification problems in the absence of noise.

- DD is exacerbated by noise (in line with Nakkiran et al. (2020a)) and its shape is affected by the pair training set density. While SNNs trained on sparse datasets show similar DD curves in the presence of SLN and PLN, these become quite distinct when the similarity relations in the training set are dense. Specifically, the interpolation threshold in the presence of PLN requires more parameters and its test error remains higher in the overparameterized region. An example of this behavior is shown in the bottom right plot of Fig. 1.

- We show that the poor performances of PLN derive from its similarity-breaking nature that manifests when input data are highly connected. We show that the interpolation threshold (training error = 0) cannot be achieved in this scenario, and we derive the analytic formula for the asymptotic training error value in the deep overparametrized regime. We call this phenomenon *Density-Induced Break of Similarity (DIBS)*.

- We test the correspondence between offline generalization and online optimization for similarity learning. We study how the architecture and the presence of noisy labels can differently impact these two regimes. We find that the conjecture only holds for clean data.

- In the presence of label noise, we find that the online/offline correspondence breaks down for all choices of training settings considered. In particular, the effect of label noise is notably more relevant in the offline case.

---

[1]Notably, it is known that the DD curve is exacerbated in the presence of random label noise in supervised classification (see, e.g., Nakkiran et al. (2020a)).

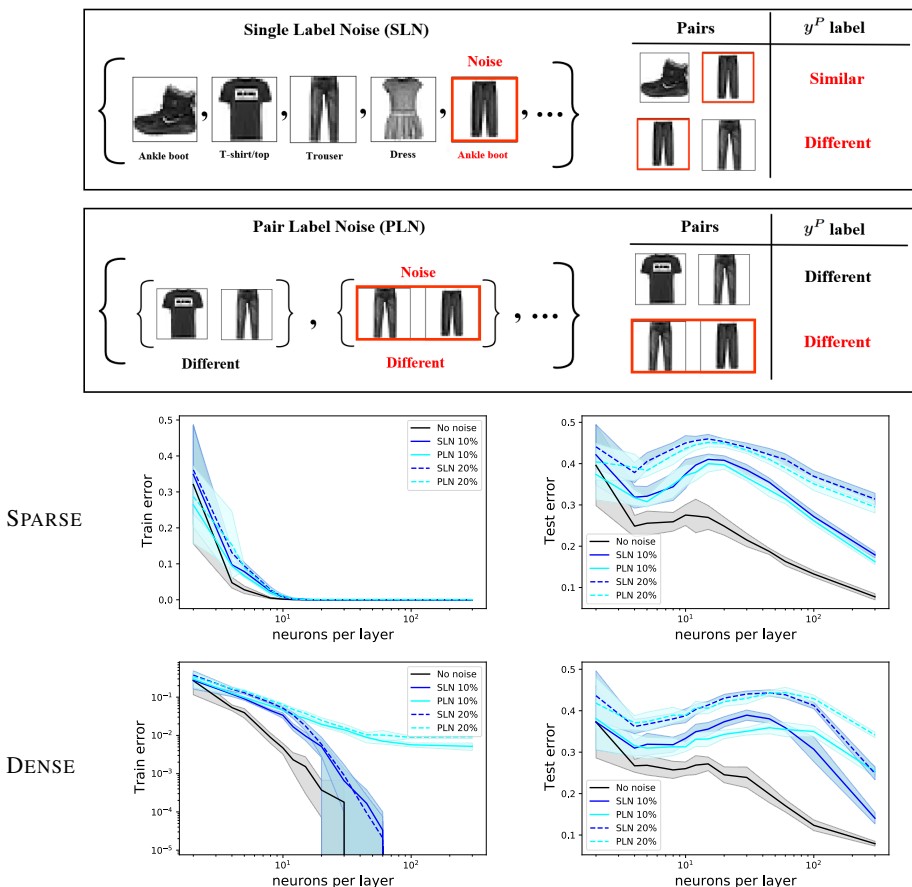

Figure 1: Top: Pictorial view of SLN and PLN in SNNs. Bottom: Train (left) and test error (right) as a function of model size. We consider a 3-layer MLP with ReLU activation function on trained on sparse and dense pairs of MNIST with 10% and 20% effective noise.

## 1.1 RELATED WORK

In the past few years, much effort has been made to understand how neural networks were able to generalize in classification problems in the presence of noise (e.g., Li et al. (2019); Han et al. (2019); Arazo et al. (2019); Harutyunyan et al. (2020); Song et al. (2020)). Remarkably, the DD behavior allowed to investigate the NN behavior as the number of trainable parameters, the evolution time and the dimensionality of the sample vary: Nakkiran et al. (2020a); Bodin & Macris (2021); Heckel & Yilmaz (2020); Pezeshki et al. (2021). Subsequently, other works have produced analytical studies of some of these phenomena: d'Ascoli et al. (2020a;b); Mei & Montanari (2022). Another complementary tool used to study generalization in classification tasks is the online/offline correspondence proposed in Nakkiran et al. (2021), which focuses on datasets without noise. This study empirically showed that a correspondence between online optimization and offline generalization holds for modern deep NNs trained to classify images. Earlier studies have proposed a similar comparison for linear models focusing on the asymptotic regime of training (see, e.g, Bottou & LeCun (2004; 2005)).

Contrastive learning, introduced by Chopra et al. (2005); Hadsell et al. (2006); Oord et al. (2018), has become one of the most prominent supervised (Khosla et al. (2020); Gunel et al. (2020)) and self-supervised (Bachman et al. (2019); Tian et al. (2020); He et al. (2020); Chen et al. (2020)) ML techniques to learn similarity relations of high-dimensional data, producing impressive results in several fields, see e.g. Le-Khac et al. (2020); Jaiswal et al. (2021). Despite its success, Ohri & Kumar (2021); Liu et al. (2021); Jaiswal et al. (2020); Le-Khac et al. (2020) show that contrastive learning usually requires huge datasets and a considerable use of data augmentation techniques. Dealing with augmentation techniques and unlabeled data where negative samples are randomly selected introduces instance discrimination challenges, i.e., the need to find ways to limit the appearance of faulty positive and negative samples. Indeed, Robinson et al. (2021); Wang & Liu (2021) show that contrastive

loss does not always sufficiently guide which features are extracted. For these reasons several works tackled the problem of discriminating against faulty negatives, as Huynh et al. (2022); Kalantidis et al. (2020); Chuang et al. (2020); Iscen et al. (2018), removing faulty positives and negatives dynamically (see Robinson et al. (2021); Zhu et al. (2021)) and creating more robust contrastive setups introducing new losses (see Chuang et al. (2022); Morgado et al. (2021)) or architectural components, Grill et al. (2020).

## 2 DATASET CONSTRUCTION

In this section we describe the choices we made to study the dataset features that influence training, i.e., the density of the image pairs and the presence of noise. We start by defining the criteria we used to construct the pairs dataset.

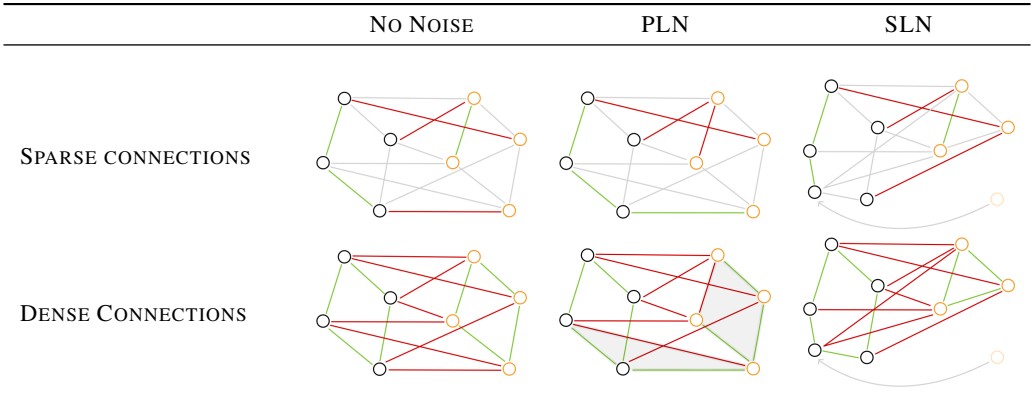

Figure 2: Pictorial view of data relation appearing in Scenario 1 (top) and 2 (bottom) for two classes of data. Positive pairs are connected by green edges, negative pairs by red edges. Ignored connections and data are in light gray. Gray-shaded areas are examples of transitivity breaking (DIBS).

**Similarity graph.** As opposed to classification problems, where the main concerns during dataset creation are class balancing and image diversity, in contrastive learning, we should consider that pair (or group) relations between images define an unoriented similarity graph inside the input space. Calling $N$ the total number of images in the full dataset, the density of this graph, $\rho = |N_{\text{pairs}}|/\binom{N}{2}$, tells us how much information we have about the input images. To maximize the information about a certain dataset, we should construct all possible labeled pairs, $\binom{N}{2} \sim N^2$, but this quickly becomes unfeasible when considering large datasets. For this reason, we construct pairs in a way that maximizes the information about similar images (all similar images are transitivity-related) and scales linearly with $N$. In practice, we construct closed chains of positive pairs within the same class, $c$, $\{\{x_1^c, x_2^c\}, \dots, \{x_k^c, x_{k+1}^c\}, \dots, \{x_n^c, x_1^c\}\}$, where $n$ is the total number of images in $c$. Then, to build negative pairs, each image is connected to a randomly chosen one belonging to a different class. If the original dataset classes are balanced, each image appears on average in 4 different pairs (2 times in the positive and 2 times in the negative pairs). Therefore, the total number of pairs is given by $N_{\text{pairs}} = 2 \times N = 2 \times n \times n_c$, where $n_c$ is the total number of classes.[2] Finally, we describe the dataset construction method we used to study how density in the similarity graph affects training.

- **Scenario 1: sparse connections.** To train the network in the absence of noise, we first create the pairs using the full dataset. We follow the procedure described at the beginning of this section so that $N_{\text{pairs}} = 2 \times N$. We then take $N_{\text{sample}}$ balanced pairs (data used to train the model) from the $N_{\text{pairs}}$ list to train the NN and repeat this procedure $n_s$ times.

- **Scenario 2: dense connections.** In this setup, we create a reduced version of the original dataset. Being interested in training the network on $N_{\text{pairs}}$ pairs, we select $N_{\text{reduced}} = N_{\text{pairs}}/2$ images from the original training set. The reduced dataset is balanced so that we have $N_{\text{pairs}}/(2n_c)$ images per class. Then, we create our training and test samples using the same prescription described at the beginning of this section. We connect adjacent images

---

[2]Note that this formula holds for dataset with at least 2 images per class.

within the same class and each of them with a random image belonging to a different class so that we get exactly $N_{\text{pairs}}$ pairs that will be automatically balanced. We repeat this procedure $n_s$ times.

Pictorial representations of the similarity graph are shown in Fig. 2, where we represent elements belonging to different classes with nodes of different colors (black and orange classes), similarity and dissimilarity relations with green and red vertices, respectively.

**Noise introduction.**    SNNs can be subjected to different types of noise having different properties. To show their impact on the training process we introduce two simple representatives, namely Single Label Noise (SLN) and Pair Label Noise (PLN) which we describe below (see Fig. 2 for an illustration).

- **Single Label Noise (SLN).** Let us consider a dataset with $N$ samples $X^S = \{x_1, x_2, \ldots, x_N\}$ belonging to $n_c$ classes and their corresponding labels $Y^S = \{y_1^S, y_2^S, \ldots, y_N^S\}$. Suppose the classes are uniformly populated. If some label noise is present in the original dataset, this will propagate to the training pairs as these are created. If SLN is uniformly introduced across all classes, it will keep the original class balancing on average (over multiple samples). On the other hand, in every single run, statistical fluctuations of uniform distribution introduce some asymmetry in the original class representative number (see Fig. 2). Finally, in SLN, similarity relations (reflexive, symmetric, and transitive properties) are preserved as mislabeling appears in all pairs containing a misclassified image.

- **Pair Label Noise (PLN).** Let us now consider a dataset of $N$ pairs $X^P = \{\{x_1^a, x_1^b\}, \ldots, \{x_j^a, x_j^b\}, \ldots, \{x_N^a, x_N^b\}\}$ with pair labels $Y^P = \{y_1^P, y_2^P, \ldots, y_N^P\}$, which can be similar ($y^P = 1$) or different ($y^P = 0$). We construct them so that they are balanced (half are similar, half different). Suppose we randomly shuffle some fraction of the total labels. In that case, the noise we introduce is symmetric under similar $\leftrightarrow$ different changes, and it acts democratically on every class of the original dataset. On the other hand, PLN can lead to inconsistent relations in the pairs dataset. Indeed, as we will show in the following sections, it breaks transitivity and, therefore, similarity.

As discussed later, these two sources of noise impact how models learn similarity relations in distinct ways. To fairly compare the outcome of the model in the presence of PLN and SLN, we need to ensure that we introduce the same amount of input label noise in the two setups. We illustrate below how we ensured that the same amount of *effective noise* was introduced. Being $n_c$ the number of image classes, $y_i^S$ the label of the i-th image, and $y_i^P$ the label of the i-th pair of images, we can define the SLN transformation as

$$\mathcal{T}_{\text{SLN}}(q) : y_i^S \to \text{random}(1, n_c) \qquad \text{with probability } q \tag{1}$$

and the PLN transformation as

$$\mathcal{T}_{\text{PLN}}(\tilde{q}) : y_i^P \to \text{random}(0, 1) \qquad \text{with probability } \tilde{q}. \tag{2}$$

As SLN appears in the dataset before pair creation and the pairs are constructed so that the dataset is balanced (half pairs are similar, half are different), the probability of effective pair mislabeling induced by SLN, $P_{\text{SLN}}(q)$, is given by

$$P_{\text{SLN}}(q) = q - \frac{q^2}{2}. \tag{3}$$

while the probability of effective pair mislabeling coming from PLN, $P_{\text{PLN}}(\tilde{q})$, is

$$P_{\text{PLN}}(\tilde{q}) = \frac{\tilde{q}}{2}. \tag{4}$$

The requirement of having the same amount of effective noise in the dataset ($P_{\text{SLN}}(q) = P_{\text{PLN}}(\tilde{q})$) boils down to the following relation between $q$ and $\tilde{q}$:

$$q = 1 - \sqrt{1 - \tilde{q}}. \tag{5}$$

The full derivation of these results and the pseudocodes describing dataset creation are given in the supplementary material in Sec. C and B, respectively.

## 2.1 EXPERIMENTAL SETUP

In this work, we consider two simple Siamese branch architectures. The first one is an MLP with 3 hidden layers having the same width and ReLU activation functions with Xavier uniform initialization, see Glorot & Bengio (2010). The second architecture is a 4-layer CNN. We also considered two training setups: in one case, we compute the Euclidean distance in the output layer training the network using Contrastive Loss from Hadsell et al. (2006), in the other one we compute the cosine similarity training the network using Cosine Embedding Loss (see Section D in the supplementary material for further details). The CNN architecture is based on the model described in Page (2018), it contains three Convolution-BatchNormalization-ReLU-MaxPooling layers and a fully-connected output layer. The number of filters in each convolution layer scales as $[k, 2k, 2k]$ while the MaxPooling is [1, 2, 8]. We fix the kernel size = 3, stride = 1 and padding = 1. When we train the network using contrastive loss (cosine embedding loss), we set the fully-connected output layer width to $k$ ($2k$).

**DD setup:** We test the presence of DD using MNIST, FMNIST (from Xiao et al. (2017)) and CIFAR10 datasets (from Krizhevsky et al. (2009)). To understand the impact of overparameterization, we study how training and test errors vary at increasing network width and training time. To do so, we increase the number of neurons per layer in the fully connected architecture and the parameter $k$ in the CNN. For all datasets, we consider 6000 training and 9000 test pairs. In every DD experiment, we let the network evolve for 2000 epochs using Adam optimizer with minibatches of size 128 and learning rate $\lambda = 10^{-4}$ except explicitly stated. All the hyper-parameters and the margins were chosen empirically. To see the average effect regardless of the particular choice of images in the dataset and weights initialization, we run 15 evolutions of the network using different training and test samples at each time. In most of the experiments, unless otherwise stated, we considered $\tilde{q} = 0.2$, i.e., an effective noise of 10%.

**Online/offline setup:** Since we cannot reuse samples for the online training, we consider an extended version of the standard MNIST dataset, namely the EMNIST (from Cohen et al. (2017)). We use the digit section of EMNIST that contains 240,000 training (and 40,000 test) $28 \times 28$ greyscale pixel images. We train Real World over 40 epochs using 12k pairs that are created considering Sparse and Dense scenarios. The Ideal World is trained once on 480k pairs created using the full training set of 240k samples. We test the models with 9k pairs constructed from the test set and consider Siamese networks with MLP and CNN blocks described in Sec. 2.1. In order to compare the results on different network architectures we used a comparable total number of parameters, namely, 200 nodes per layer for the MLP cases (total of 237,400 parameters) trained with the contrastive loss ($\lambda = 10^{-4}$); and width $k = 47$ (total of 235,611 parameters) for the CNN cases trained with the cosine loss ($\lambda = 5 \times 10^{-5}$). To provide an estimate of the results regardless of the particular choice of images and network initialization, we run MLP (CNN) experiments 5 (4) times.

All our experiments make use of the TensorFlow/Keras framework: Abadi et al. (2015). Each of the experiments mentioned above was performed in the presence and absence of noise and considering sparse (scenario 1) and dense pairs (scenario 2) in the training set.

## 3 RESULTS

**DD results.** In all experiments, we can see the parameter-wise DD, regardless of architecture, loss function, scenario and noise. To support the consistency of our results with previous DD literature, in the supplementary material we also investigate the presence of epochwise DD in SNNs. Both kinds of DD are observed under every experimental condition. This does not happen in classification problems which typically require the presence of noise to make the DDs clearly visible (see, for example, Nakkiran et al. (2020a)). As expected, DD becomes more prominent in the presence of noise. In Fig. 1 we show how the network reacts to different amounts of noisy labels. Our results, showing the average training and test errors together with error bars, can be found in Fig. 1 and in the supplementary material in Sec. G.1. In **Scenario 1**, the input dataset connections are sparse, and PLN and SLN have the same impact on training. This makes sense as there should not be any difference between PLN and SLN effects in the extreme case where every image appears only once in the training set. Instead, **Scenario 2** is characterized by dense input connections, and the system behaves differently under SLN and PLN. We experimentally observe that the DD peak location changes in some but not all setups. Specifically, this happens in MLP-Euclidean Distance and CNN-Cosine Loss setups where the PLN peaks appear to be shifted to the right-hand side, hinting that PLN noise is harder to interpolate than SLN. No such thing appears using CNN-Euclidean Distance.

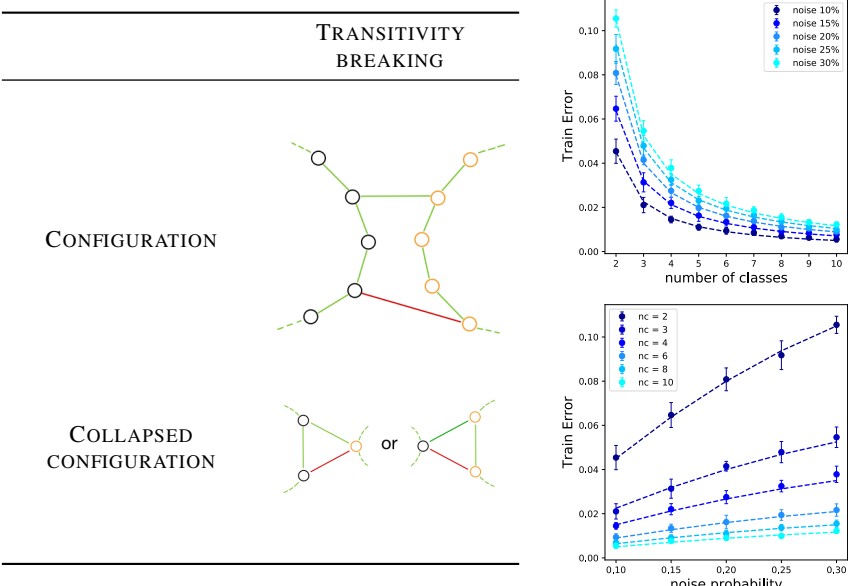

Figure 3: Left: leading transitivity breaking configuration (top) and its collapsed versions (bottom). Right: Analytic (lines) and numerical (scatter points) estimates of the asymptotic training error behavior at varying number of classes $n_c$ (top) and effective noise (bottom) in the presence of PLN in scenario 2.

We believe that the presence of this shift signifies that the right NN setup is being used (i.e., the natural architecture-loss function match).[3] Increasing the amount of noise enhances the test errors as expected, but does not induce any significant peak shift. SLN test error tends to be higher in small to medium network sizes. A hint about how this happens is given in Fig. (2). Indeed, SLN introduces a systematic error: a mislabeled image appears to be mislabeled in every pair. Therefore, given that the image features are not going to agree with pair labels, the only way the network has to classify correctly is by extracting the image from its natural distribution. NNs being continuous functions, this implies that a neighborhood of said image must be extracted as well, increasing the test error. At higher network widths, the volume of the mislabeled image neighborhood can become arbitrarily small, and the test error is free to go down again. In fact, *SLN introduces systematic errors that do not compromise the consistency of the similarity graph*. On the other hand, PLN stays higher in the deep overparametrized regime. Indeed, randomly changing similarity relations in the input dataset, *PLN ends up breaking transitivity, making the training set similarity graph inconsistent*. Beyond keeping test error high, this inconsistency also implies that the network will never be able to overfit completely: the training error will no longer vanish just by increasing the number of network parameters.

**Origin and magnitude of DIBS.** We now explain the origin of the phenomenon we call *Density-Induced Break of Similarity* originating from PLN. A similarity relation must satisfy transitivity. We can see if transitivity is satisfied or violated in the training set by evaluating the consistency of the closed paths in the similarity graph. See for example Fig. (2), where we highlight some inconsistent paths with gray areas. To facilitate this operation in more complex setups, we can also resort to collapsed configurations, i.e., collapsing nodes connected by green vertices (see discussion around Fig. 8 in the supplementary material). The analysis of the collapsed configurations shows that transitivity breaking mainly derives from the configuration in Fig. 3. The asymptotic training error (at leading order) is given by the probability associated to this local triangular configuration, given the dataset construction method explained in Sec. 2. This is given by the probability of having a misclassified equal pair, attached to a correctly classified one. Both pairs should be connected to the same other class of images. Finally, we need to take into account the number of possible configurations. These are two and are given by swapping colors between the two pairs attached to

---

[3]This intuition is supported by further online/offline correspondence experiments we present in the supplementary material.

different classes. This leads to:

$$\underbrace{\frac{N_{\text{eq.pairs}}}{N_{\text{pairs}}} \times P}_{\text{misclassified equal pair}} \overbrace{\frac{N_{\text{diff.pairs}}}{N_{\text{pairs}}} \times (1 - P)}^{\text{correct different pair}} \underbrace{\overbrace{\frac{2}{n_c - 1}}^{\text{\# configurations}}}_{\text{connected to same 2 classes}}$$

where $N_{\text{eq.pairs}}$ refers to the number of equal pairs (similar), $N_{\text{diff.pairs}}$ is the number of different pairs (dissimilar), and $N_{\text{pairs}}$ is the total number of pairs, with $\frac{N_{\text{eq.pairs}}}{N_{\text{pairs}}} = \frac{N_{\text{diff.pairs}}}{N_{\text{pairs}}} = \frac{1}{2}$ as we consider balanced pairs. Therefore, the dominant contribution to the asymptotic training error is given by:

$$\lim_{n_\theta \to \infty} \text{TrainError}_{\text{Dense}}^{\text{PLN}}(P, n_c) = \frac{P(1 - P)}{2(n_c - 1)}, \tag{6}$$

where $n_\theta$ is the number of network parameters and $P = P_{\text{PLN}}(\tilde{q})$ is the effective amount of noise. In Fig 3, we validate our formula by comparing it with experimental results. There, we consider the FMNIST dataset trained on our MLP architecture with 500 neurons per layer, using Euclidean distance and contrastive loss. Numerical results (mean and standard error bar) come from 10 runs where we choose different random classes each time. These results show how, in the overparametrized regime, the training error changes with the effective noise and with the different number of classes. This analysis shows that the macroscopic presence of transitivity breaking is linked to the presence and number of closed paths in the similarity graph and therefore to the dataset density.

**Online *vs.* offline learning.** We probe the correspondence between offline generalization and online optimization (Nakkiran et al. (2021), see Sec. F for details) for similarity tasks by studying how the training setting and the presence of noisy labels can impact these two regimes. Considering usual training settings (i.e., natural choices of architecture-loss function match), the conjecture holds for data without noise, regardless of the dataset density. In the presence of label noise, however, we find that the online/offline correspondence breaks down for all choices of training settings considered.

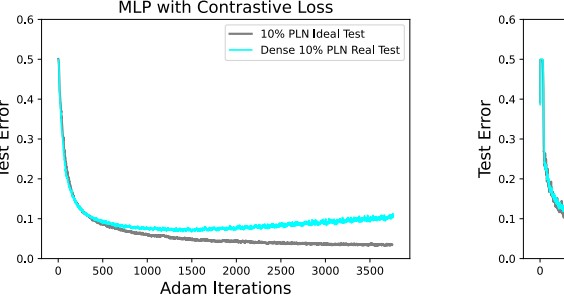
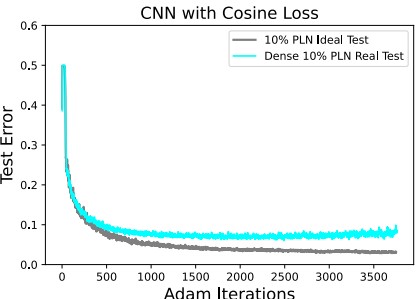

Figure 4: Ideal *vs.* Dense Real worlds with 10% PLN. Plots show the Test Errors as a function of minibatch Adam iterations for a Siamese architecture with MLP (left) branches with 200 nodes per layer and with CNN (right) blocks with width $k = 47$. The architectures details are given in Sec. 2.1.

Two representative examples where the conjecture breaks are depicted in Fig. 4. There, we show the median test error values on dense dataset of real- and ideal-world scenarios with 10% of PLN trained using MLP (left) and CNN architecture (right). We compare offline and online settings after the same number of training iterations. We observe that while both Ideal and Real test errors are affected by noise, this effect is exacerbated in the Real World scenarios. This can be understood because "fresh" samples bring more diversity to the model, improving generalization even if these new samples have noisy labels. Interestingly, we find that the online/offline correspondence for similarity tasks is influenced by the network architecture and the loss function choice. In particular, Fig. 4 shows that the Real-world scenario for the MLP architecture diverges from the corresponding online case earlier (with less iteration steps) than the CNN case. Nevertheless, independently of the architecture-loss matching, the equivalence between online and offline settings breaks down in the presence of label noise for all the scenarios considered. Similar behavior occurs for the sparse case as shown in Sec. G.5 in the supplementary material. There, we also present several additional comparisons between the architectures, losses and noise levels.

**DIBS and modern contrastive learning.** The similarity-breaking nature of PLN in dense datasets should not be underestimated as it may appear in widely employed settings. Modern approaches to self-supervised contrastive learning (see the recent reviews of Ohri & Kumar (2021); Liu et al. (2021); Jaiswal et al. (2020); Le-Khac et al. (2020)) heavily rely on data augmentation to learn representations, Tian et al. (2020). The massive use of data augmentation, however, may result in partial representation learning (feature suppression) or lead to semantic errors as in Purushwalkam & Gupta (2020). Moreover, as exposed in Huynh et al. (2022), if negative pairs are formed by sampling views from different images, regardless of their semantic information, this may lead to the appearance of false-negative pairs, potentially breaking transitivity and compromising the training efficiency. Interestingly, this skewness towards false-negative pairs is the same effect we find studying the asymptotic training error balance with DIBS (see discussion around Fig. 7 in the supplementary material). Despite these problems, data augmentation and random selection of negative samples are intrinsic to self-supervised methods.[4] Therefore, several works in contrastive learning have focused on controlling the quality of augmented data and mitigating the effects of false negatives[5] (see section 1.1). For this reason, feature extraction in self-supervised contrastive learning is usually affected by pair label noise by construction.

## 4 DISCUSSION AND CONCLUSIONS

We move the first steps towards understanding generalization in similarity learning focusing on SNNs. To do so, we borrow the frameworks of DD and online/offline correspondence from classification tasks. We show that DD appearance is magnified in SNNs as it appears also in the absence of noise. Notably, we find that noise and the density of pairs in the training set crucially affect generalization. We present two kinds of noise: SLN, preserving similarity relations, and PLN, breaking transitivity. The same noise sources presented in this work can be easily generalized to models where the network input is given by multiple images. Studying DD, we show that similarity-breaking noise compromises the asymptotic generalization performance (large training time) of the network in the overparametrized regime. Moreover, these effects get magnified at increasing training set density, preventing perfect interpolation. Studying the online/offline correspondence, we find that the generalization properties before overfitting time are not sensitive to the density of the training set and only depend on noise. In particular, in the presence of noise the online/offline correspondence breaks down and the differences between the real and ideal generalization gap are not universal and depend on the training setup.

**Limitations.** This is an exploratory work that does not investigate all possible setups which may affect or lead to DD, such as regularization (see Nakkiran et al. (2020b); Mei & Montanari (2022)), epoch and sample-wise DD (see, Nakkiran et al. (2020a); Bodin & Macris (2021); Heckel & Yilmaz (2020); Pezeshki et al. (2021)). Moreover, we focus on the under- and over-parametrised regime without providing quantitative results about the interpolation threshold itself, d'Ascoli et al. (2020a;b); Mei & Montanari (2022). This is because, to the best of our knowledge, there is no predefined way of treating SNNs analytically as no proxy model as Random Fourier Features (see Rahimi & Recht (2007)) can be constructed. Indeed, while in classification or regression tasks the output layer size is known, this is not true for SNNs. For this reason, we believe that an analytic study of DD in SNNs may require some other approach, and we leave this study for future work.

**Outlook.** In the majority of modern contrastive learning works, the final graph of similarity relations in the dataset becomes really dense as each training step involves multiple images at a time. Moreover, from instance discrimination task examples, we know that contrastive learning tends to be affected by faulty positive and negative pair relations. This is the setting were we find that noise crucially impacts generalization. While the technological developments and the applications of contrastive learning kept on expanding during the last years, a fundamental study about how it generalises and reacts to noise is still missing.

---

[4]For example, in a pretext task, the original image acts as an anchor, its augmentations act as positive samples, and the rest of the images in the batch (or in the training data) act as negative samples.

[5]Indeed, when two different images belonging to the same class of objects (sharing semantic features) are classified as different, convergence slows-down and semantic information gets lost. This goes under the name of instance discrimination task (i.e., the problem of discriminating pairs of similar points from dissimilar ones), and failing it can be harmful to the formation of features useful for downstream tasks.

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
