# OpenReview forum: "Similarity and Generalization: from Noise to Corruption"
_ICLR.cc/2023/Conference — Submitted to ICLR 2023_

### Official Review · Reviewer_JwMY · 2022-10-14

**Confidence:** 3
**Correctness:** 2
**Technical Novelty And Significance:** 3
**Empirical Novelty And Significance:** 3
**Recommendation:** 5

**Clarity, Quality, Novelty And Reproducibility:**

Here, I make a list of unclear points in the paper. I hope this helps the authors to improve the manuscript.

- (In the abstract, introduction, and several places) The authors say "NNs generalize the concept of similarity" occasionally, which is confusing because the verb "generalize" in the machine learning context should be an intransitive verb. I guess an alternative way to express what the authors initially thought could be "NNs generalize in similarity learning," which causes less misunderstanding.
- (In the abstract) The authors state "SLN outperforms PLN in the overparametrized region," which sounds like the authors propose a new noise model to achieve better performance by injecting noises on purpose, causing a misunderstanding regarding the aim of this paper.
- (In the introduction; 1st paragraph) The connection between the online-offline learning correspondence and double descent is not sufficiently understandable from the current sentences. In particular, the last part "overparameterized models (trained on a finite number of samples) and underparameterized models (trained on very large datasets)" would be ambiguous. In addition, I would appreciate if the authors state slightly more about why the topic drifts to the online/offline correspondence from the double descent in the middle of this paragraph.
- (At beginning of page 2) The sentence "online/offline training compares the network performances varying the quality (diversity) of the dataset" puzzles me. What is the quality (diversity) of the dataset in this context and what is the related concept in the latter part of this paper? Does online/offline training itself vary the quality?
- (At the bottom of page 3) I failed to understand what the phrase "dataset quality and pairs topology" indicates. Does the "pairs topology" refer to SLN vs. PLN?
- (In Section 2) In the data-generating process introduced in the 2nd paragraph of Section 2, the authors choose to generate positive pairs within the same class in a "chain" manner, but why is it? It does not seem to represent a realistic data-generating scenario.
- (In Section 2.1) What is the purpose to introduce two scenarios, sparse and dense connections? It looks like the authors intend to accelerate data sampling with these two new scenarios, but do they represent a realistic scenario? I am not sure whether they are just a proxy model to a real scenario or a proposed sampling method for some goals.
- (In Section 3) The last paragraph of page 7 may be excessively long. It can be split into several pieces because it contains several important topics.
- (In Section 3; the last paragraph of page 7) In the last part, the authors claim "we need to take into account the number of possible configurations" but I failed to understand what is "possible configurations" and for what purpose we *need* to think about it. In addition, I am not sure what the last equation shown in page 7 calculates (partly because of some undefined notations like $N\_{\\mathrm{eq.pairs}}$).
- (In page 8; the latter part of the 1st paragraph in "Online vs. offline") Could you explicate why an online/offline correspondence is supported by *showing that the bootstrap error is small*?

I feel slightly sorry for pointing out structural and grammatical issues too much but could not avoid doing so because these issues hinder me from understanding the main scope of this work. I believe improving these aspects must make the paper more attractive.

In addition, while looking at the experimental results, I am not confident enough that "the PLN peaks appear to be shifted to the right-hand side, hitting that PLN data are harder to interpolate" (stated in the middle of page 7) from Figure 1 (bottom right). Both peaks of SLN and PLN range over similar regions and the difference does not look significant.

**Strength And Weaknesses:**

### Strengths

This work extends the concept of double descent from well-studied classification and regression to similarity learning. Many recent deep learning training procedures partly rely on self-supervised pre-training models that leverage contrastive learning, hence it is essential to understand more what kind of regime goes on under the over-parametrized regime in similarity learning. The authors provide one perspective to this answer to reveal that pair label noise has a severe effect on the generalization aspect. This should be an interesting perspective for researchers studying generalization.

### Weaknesses

The main drawback of this paper is the unclearness of the motivation and writing. I provide several writings that could be improved in the "Clarity, Quality, Novelty and Reproducibility" section and focus on the motivation aspect here.

Although research on double descent/interpolation/over-parametrization regime has been gaining attention in the recent community, I could not understand well what is the motivation of this work. In the study on double descent, we are primarily interested in how model performances degrade and improve as the model size increases, and there should not be label noises. Specifically, we usually consider the data-generating process $Y_i = f(X_i) + \\epsilon_i$ for regression and treat $\\epsilon_i$ as "white noise," but do not consider "more drastic (label flipping) noise" to purely focus on the clean generalization aspect. The current study seems to entangle labeling noise and the generalization aspect, so it is not sufficiently clear that the authors would like to focus on generalization or the effect of the noise.

In addition, the authors introduce two types of noise models (SLN and PLN) and two scenarios (sparse and dense connections). However, I did not understand well the reason why the authors introduce them. Are these introduced to imitate the real-world data-generating process in similarity learning? Or, are these introduced to make the data-generating process more scalable (in particular, the sparse connection)? This part makes me puzzled so I'm not sure the authors would like to study/reveal a certain phenomenon or propose a new data sampling procedure.

As for SLN and PLN, if we follow the initial motivation---to study the generalization aspect of contrastive learning with data augmentation and negative sampling---then these noise-generating processes could be overly simple. At least, in real-world scenarios, SLN and PLN could happen simultaneously. Why do we consider these two noises independently?

**Summary Of The Paper:**

This paper empirically investigates the phenomena of double descent and offline-online correspondence (the gap between test errors of offline and online learners, as far as I understood) for similarity learning. The authors introduce two types of dataset conversion procedures from data points to pairs and two types of noise-generating processes to see how these factors affect generalization in similarity learning. These data-generating processes are expected to imitate well actual procedures in contrastive learning with data augmentation and negative sampling. After all, the authors discover for double descent that a type of noise---called pair label noise (PLN)---affects generalization more significantly because it breaks the label "transitivity," making the interpolation/overfitting harder. In addition, the authors reveal that offline learners are more prone to label noise than online learners in the empirical study of offline-online correspondence.

**Summary Of The Review:**

I think this paper reveals an interesting contrast between the behavior of SLN and PLN in regard to the double descent phenomena. Nevertheless, several parts of the paper would not be sufficiently clear so the scope of the paper is not easy to grasp. There would be room for improvement in the structure and logical flow of the paper as suggested in "Clarity, Quality, Novelty And Reproducibility" section.

---

> ### Author Response · Authors · 2022-11-18
> **Reply to reviewer JwMY**
>
> First, we want to thank the reviewer for his/her thoughtful comments and suggestions for improving the quality of our work. As suggested, we have reorganized the paper's content and figures to make the motivation and results more transparent and easier to read.
>
> About weaknesses:
>
> Let us first clarify the relation between noise and the double descent (DD) phenomenon. We agree with the reviewer that there is a vast literature studying DD in regression, where white noise is inherent to the data-generation process. On the other hand, it is known that the DD phenomenon is exacerbated by label noise in classification tasks, i.e., label flipping (see, e.g., "Deep Double Descent: Where Bigger Models and More Data Hurt", Nakkiran et al., ICML 2020 ). Oppositely to classification problems, similarity tasks allowed us to explore noise sources with different properties: PLN and SLN. Interestingly, these two noise sources not only exacerbate the DD curve but give rise to completely different learning models in the over-parameterized regime (as observed in all our DD plots for densely connected datasets). As discussed in the paper, PLN similarity violation becomes macroscopical, corrupting the dataset to the point where complete overfitting cannot be achieved. We call this phenomenon Density-Induced Break of Similarity (DIBS). To our knowledge, DD was never studied in the context of similarity learning. Therefore, studying how SNNs react to noise in the under and over-parameterized regimes enabled us to observe DD in similarity learning and also allowed us to find a new phenomenon (DIBS) that does not appear in classification problems. We thank the reviewer for pointing out that this needed to be clarified.
>
> To summarize, our intention is not to propose a new data sampling procedure but to study/reveal a new phenomenon. We agree with the reviewer that in real-world scenarios, PLN and SLN could happen simultaneously; however, exploring this effect is out of the scope of this paper. Indeed, here we are not aiming to cover an exhaustive list of label noise combinations, which can be an interesting follow-up work.
> Here we want to explore the simplest possible scenarios where similarity preserving or breaking noise appears.
>
> About: quality (diversity) of the dataset and online/offline correspondence?
>
> The study about online/offline training compares the test errors between ideal (online) and real (offline) worlds. Since both settings are trained for the same number of optimizer steps on the same architecture, the difference between these setups is given by their data imbalance: the ideal world only sees fresh samples, real-world sees a fixed number of data over many epochs.
>
> About: Meaning of dataset topology
>
> By "pairs topology," we meant the sparse vs. dense connections. This sentence has been changed for clarity.
>
>
> About: Chain configuration and the purpose of sparse and dense connections.
>
> Studying the densities of the pairs in the dataset is essential as it impacts the learned model. In particular, the datasets used in SNNs tend to be dense. The chain configuration is one of several ways to create pairs; another is to create them entirely randomly (with the risk of inserting multiple identical pairs). Even in modern contrastive learning, the simultaneous processing of multiple images (triplets, multiplets, or the entire dataset) gives rise to dense similarity graphs (see paragraph on modern contrastive learning in Results). We have shown how that training on sparse datasets hides the existence of the similarity graph (which is disjoint), and all noises have the same effect. Training on dense datasets reveals the importance of the similarity graph and its possible inconsistencies due to noise. We believe that changing the pair creation procedure would not lead to qualitatively different results as long as it allows for "triangle" configurations, as shown in Figs. 2 and 3.
>
>
> About: online/offline correspondence and bootstrap error
>
> See answer 7. to Reviewer egc1.
>
>
> About: the PLN peaks appear to be shifted to the right-hand side.
>
> We agree with the reviewer that the peak shift effect is not systematic. As we explain in the DD section of the results (see new draft), the shift depends on the characteristics of the learning model. However, in all cases where we train SNNs using (Euclidean distance + contrastive loss) or (cosine similarity + cosine loss), we see the shift.
> See bottom right plots in Figs. 1,10,13,14,16.
> This tells us that PLN is generally more complex to interpolate.
> In order to better see this effect in Figure 10, we increased the number of runs to 20 and replaced the image.

---

> > ### Comment · Reviewer_JwMY · 2022-11-25
> > **Reply**
> >
> > I thank the authors for addressing the comments dedicatedly. The motivation part has been made much clearer in the response and hence I raised the score. I agree that we need not consider the complicated combination of PLN and SLN for the first step.

---

### Official Review · Reviewer_egc1 · 2022-10-17

**Confidence:** 4
**Correctness:** 3
**Technical Novelty And Significance:** 2
**Empirical Novelty And Significance:** 3
**Recommendation:** 5

**Clarity, Quality, Novelty And Reproducibility:**

Clarity. At the present stage, there are some concerns that need to be addressed:
- The paper claims that "SLN outperforms PLN in the overparametrized region in dense datasets". Both SLN and PLN are noise. Does the paper mean that the SLN noise is less harmful in this case?
- For claims "online and ofﬂine test soft-errors match each other for classiﬁcation tasks under certain conditions", is there some evidence for this statement?
- Perhaps, it is better to stress similarity learning but not contrastive learning, since the paper mainly exploits SNNs.
- What are "reﬂexive, symmetric, and transitive properties"? Could the paper provide more formal definitions for these properties?
- The paper shows that the interpolation threshold (training error = 0) cannot be achieved in some cases. Is this because the network capacity is not large enough, because prior works, e.g., [1] show that deep networks can fully fit any given noisy labels?
- The double descent phenomenon in learning with label (label-pair) noise is not very exciting. In fact, prior works such as [2] have presented this phenomenon through training dynamics. Intuitively, since clean labels are dominant in noisy classes, the double descent phenomenon exists when we train deep networks on noisy data.
- Could the paper provide more discussions about why an online/offline correspondence is supported by showing that the bootstrap error is small? It is confusing to me.

Quality. The quality of this paper should be improved, although the presentations in the experimental parts are admirable.

Novelty. The conceptual and technical novelty is not prominent.

Reproducibility. The reproducibility is good. The paper provides detailed descriptions of implementations.

**Strength And Weaknesses:**

**Strengths**
- The research problem is interesting and significant.
- Experiments are sufficient to support claims.

**Weaknesses**
- The main contributions of this paper are unclear.
- Technical novelty is somewhat limited.
- Both writing and organization should be improved to reach the requirements of a top-tier conference.



**Summary Of The Paper:**

This paper studies how deep networks generalize the concept of similarity in the presence of noise. Specifically, two phenomena are studied (1) double descent behavior and (2) online/offline relations. The double descent behavior also exhibits in the cases of contrastive learning with noise. Besides, the equivalence between online optimization and offline generalization is probed. An extensive empirical study is provided to justify the paper's claims.

**Summary Of The Review:**

This paper studies an important problem. Extensive empirical evidence is provided. However, at the present stage, there are still some unclear explanations that need to be addressed. Therefore, the rating is "5" before the rebuttal.

---

> ### Author Response · Authors · 2022-11-18
> **Reply to reviewer egc1**
>
> First, we want to thank the reviewer for his/her thoughtful comments and suggestions for improving the quality of our work. As suggested, we have reorganized the paper's content and figures to make the motivation and results more transparent and easier to read.
>
> 1. We meant that SLN is less harmful in the overparameterized region, as suggested by the reviewer. We agree with the reviewer that the sentence was unclear. This has been fixed in the new version.
>
> 2. Here, we are referring to the original paper proposing the online/offline correspondence for classification tasks (Nakkiran et al. (2021)), which we cited a few lines above this sentence. This sentence was rephrased in the new version.
>
> 3. In the abstract, we say that we focus on the simplest contrastive learning case, represented by Siamese NNs, which is an architecture designed to detect similarity relations. We stress this in the new introduction.
>
> 4. reﬂexivity, symmetry, and transitivity are the defining properties of similarity. Applying them to images we have:
>  - reﬂexivity: every image is similar to itself
>
>  - symmetry: if image A is similar to image B, then image B is similar to image A.
>
>  - transitivity: if image A is similar to image B and image B is similar to image C, then image A is similar to image C
>
>
> 5. This is a crucial point as it is related to one of the main results of our work. In fact, unlike in classification with noisy labels, we show that the interpolation threshold cannot be achieved in some cases in similarity learning due to what we call DIBS (Density-Induced Break of Similarity). In this case, zero training error cannot be achieved even if the number of parameters in the model is far above the number of training samples (i.e., regardless of the network capacity). We have analytically obtained the dominant contribution of the asymptotic train error for the Pair-Label-Noise case, which is given in Equation 6. This analytical expression is validated by the experiments in Figure 3 (right). We thank the reviewer for pointing out that this was unclear. Given the relevance of this question, we rewrote part of the introduction to emphasize this point.
>
>
> 6. We are unsure about what is [2] (It does not seem that the reviewer is referring to Huang et al. 2019. (second paper cited in our work) or Adlam et al. 2020 (the second reference appearing in the bibliography). Could you please specify what reference is [2]?
> Nevertheless, to our knowledge, the double descent phenomenon was never studied, observed, or analyzed in the context of similarity learning in the previous literature. We acknowledge that there is a vast literature discussing single-label noise for classification. However, here we point out that noise in SNNs may show very different properties. In fact, any noise operating on single image labels cannot induce transitivity breaking, which is the main peculiarity of pair-label noise.
>
> 7. If the bootstrap error is small, we expect that models with good optimization in the infinite-data regime (online, i.e., 'ideal world')  also generalize well in the finite-sample regime (offline, i.e., 'real world'). Note that this connection is non-trivial: the infinite-data limit sits in the under-parameterized region (number of samples $>$ number of parameters), while the finite-data case corresponds to the overparameterized regime for modern deep neural networks (number of samples $<$ number of parameters).
> Concretely, the online/offline correspondence holds if the test error in the ideal world (i.e., the model only sees fresh samples) is close to the test error in the real world (i.e., the model sees the same data over many epochs). To fairly compare the results, the same number of optimizer iterations should train the two scenarios. The difference between these test errors is the bootstrap error, which should be small for the online/offline conjecture to be valid.

---

### Official Review · Reviewer_UiBx · 2022-10-24

**Confidence:** 3
**Correctness:** 2
**Technical Novelty And Significance:** 2
**Empirical Novelty And Significance:** 2
**Recommendation:** 5

**Clarity, Quality, Novelty And Reproducibility:**

Clarity: Overall this paper is not well-written. The introduction section makes up half of the whole article, which is too long. The position of the figures is not well designed. I don't see why figures of totally different meanings are combined together. As pointed out above, many sentences are hard to read.

Quality: This work is not theoretically evaluated. The empirical study is a little bit rough.

Novelty: The main claim, i.e., the noise affects the generalization of similarity learning seems to be obvious.


**Strength And Weaknesses:**

This work studies how NNs generalize the concept of similarity in the presence of noise by investigating two phenomena: Double Descent (DD) behavior and online/offline correspondence, which is interesting.

Below are some concerns:
1. The authors stated that it is widely known that larger models (with more parameters) usually obtain better generalization. However, in my point of view, over-parameterized models usually have the overfitting problem, leading to bad generalization ability.
2. What are the limits in `this framework connects under- and overparameterized limits.'
3. I am confused about this sentence `the training dataset size dictates the two regimes instead of the number of parameters'. Do you mean ``the training dataset size instead of the number of parameters dictates the two regimes''?
4. What do you mean by ``online and offline test soft-errors match each other ''?
5. I don't see why DD and online/offline correspondence are two **complementary** approaches in terms of generalization. Can you provide some explanation on the **complementary**?
6. ``DD and online/offline correspondence were mainly applied to classification and regression, but, if true, they should also hold for other tasks such as similarity learning.''. What do you mean by `if true'?
7. What is DIBS? What do the gray shaded areas stand?
8. So many special terms are mentioned without definition, e.g., effective noise, which make it hard for readers to understand this paper.
9.  The density of this graph is defined as $\left|N_{\text {pairs }}\right| /\left(\begin{array}{c}
N \\
2
\end{array}\right)$ ", which tells us how much information we have about the input images. The density is proportional to $1/N$. How do you maximize the information by constructing all possible pairs (increasing N)?
10. In scenario 1, how do you determine $N_{sample}$?
11. Can you provide some theoretical analysis on the generalization in similarity learning with two kinds of label noise under two scenarios in terms of DD and online/offline correspondence?

**Summary Of The Paper:**

This paper studies generalization in similarity learning with noisy labeled data focusing on SNNs. Empirical studies were designed to show that dataset topology crucially affects generalization under SLN and PLN, and both the network architecture and the loss function choice can disturb an online/offline correspondence for similarity tasks.

**Summary Of The Review:**

To sum up, the contributions of this paper are limited. It would be better if the authors can provide some theoretical analysis and refine the writing of this work.

---

> ### Author Response · Authors · 2022-11-18
> **Reply to referee UiBx**
>
> First, we want to thank the reviewer for his/her thoughtful comments and suggestions for improving the quality of our work. As suggested, we have reorganized the paper’s content and figures to make the motivation and results more transparent and easier to read.
>
> 1. We stress that our observation in this sentence is in the context of modern neural networks. In fact, there is overwhelming evidence among ML practitioners that larger models usually generalize better (see, e.g., Szegedy et al. (2015); Huang et al. (2019); Radford et al. (2019)). We agree with the reviewer that this is intriguing. Indeed, the double-descent work (Belkin et al. (2019)) empirically proposes a way to reconcile conventional wisdom in classical statistical learning with ML practitioners’ observations.
>
> 2. The under-parameterized limit refers to $N_{\textrm{samp.}}/N_{\textrm{param.}} \gg 1$, and the overparameterized limit refers to $N_{\textrm{samp.}}/N_{\textrm{param.}} \ll 1$, where $N_{\textrm{samp.}}$ is the number of samples in the training set, and  $N_{\textrm{param.}}$ is the number of parameters in the model.
> The transition between these regions is named the interpolation threshold. Intuitively, if the model sits in the region above the interpolation threshold, its ‘effective complexity’ is large enough to interpolate the training set (Nakkiran et al. (2020a)). For simple cases such as regression, the transition between these regions can be exactly determined and is given by $N_{\textrm{samp.}}= N_{\textrm{param.}}$ (see, e.g., Belkin et al. (2019)).
>
> 3. Thanks for pointing this out. Your interpretation was correct.
>
> 4. We mean that the offline and online soft-error are approximately equal after training their corresponding settings by the same number of iterations. This has been clarified in the new version.
>
> 5. The model-wise DD curves show how the generalization error varies with the number of parameters in the model and evaluates it at training convergence (after a large number of epochs). The online/offline comparison shows how the generalization error depends on the dataset’s quality and evaluates it before overfitting starts. Both approaches are used to study generalization.
>
> 6. Although the DD phenomenon and the validity of the online/offline correspondence have mainly been studied on classification and regression problems if they are associated with some fundamental properties of DNNs, they should also hold for other tasks such as similarity learning. They should be independent of the artifacts of the model, the specific learning task, or the optimization procedure.
>
> 7. DIBS (Density-Induced Break of Similarity) is the phenomenon induced by the transitivity-breaking nature of PLN. It is the main reason why the network cannot perfectly interpolate the training data, regardless of the NN size. We rearranged and highlighted the discussion about DIBS in the new draft (introduction and Section 3). The gray DIBS regions highlight the transitivity-breaking paths in the dataset of pairs. For example, note the gray triangle area for the PLN case with dense connections (left panel of Figure 2), where an element of the orange class forms both a positive and a negative pair with two other elements of the black class, which breaks transitivity.
>
> 8. The term 'effective noise' was described in Sec. 2.2 Effective Noise (now it was moved to Sec.2 DATASET CONSTRUCTION). Summarizing: the effective noise is given by the average value of the percentage of training pairs classified incorrectly.
>
> 9. Let us fix the dataset size to $N$. The density of the similarity graph is $|N_{\rm pairs}|/\binom{N}{2}=\frac{2 |N_{\rm pairs}|}{N(N-1)}$. If we construct all possible pairs: $N_{\rm pairs}=\binom{N}{2}=\frac{N(N-1)}{2}\sim N^2$ and the density of this graph is 1 (its maximum value). This maximizes the dataset information to be injected into the SNN problem by simply creating all possible pairs from the original image datasets. This is intractable for large datasets and contains several redundant relations.
>
> 10. In the description of scenario 1, $N_{\textrm{sample}}$ refers to the number of pairs used to train the model (the training set size). In particular, in the DD plots, all the models (both scenarios) are trained with $N_{\textrm{sample}}=$6k pairs. The notation has been clarified.
>
> 11. Studies of DD and online/offline correspondence are typically empirical. In fact, the interpolation threshold in parameter-wise DDs was only quantified on simple models and in the context of regression/classification problems (e.g., using Random Features models). This result is not readily extendable to SNNs since, in this case, the dimensionality of the output is not fixed a priori (unlike in regression or classification problems). In this work, we provide an analytical derivation of the magnitude of the asymptotic test error in the presence of PLN.
>
> Points 2,3,4,5,6,7,8,10 are clarified and extended in the new version of the draft.

---

> > ### Comment · Reviewer_UiBx · 2022-11-29
> > **Reply**
> >
> > I thank the authors for addressing my concerns. The current version is easier to read hence I raised the score.

---

### Decision · Program_Chairs · 2023-01-20

**Decision:**

Reject

**Justification For Why Not Higher Score:**

This paper provides an interesting perspective to understand similarity learning and generalization. Empirical results are provided to support its claims. Before rebuttal, reviewers raise several concerns about this paper. The authors provide responses to them.

The responses address partial concerns. Reviewers hence become more positive about this paper. However, they still think that this paper cannot reach the acceptance line, particularly due to the unclear motivation and writing that should be promoted. Minor modifications are not enough for acceptance. AC checks the opinions of reviewers and this paper, and agrees with reviewers. Therefore, the recommendation is "reject".

**Justification For Why Not Lower Score:**

N/A

**Metareview: Summary, Strengths And Weaknesses:**

**Summary**

 This paper presents work to investigate the double-descent phenomena and offline-online correspondence. Two kinds of data transform from individual points to pairs and two noise-generating processes are introduced to see how the factors influence generalization. Empirical studies are provided to show that pair-label noise can influence generalization more significantly. Also, offline learners are more prone to label noise than online learners.

**Strengths**
- The research problem is interesting and important. Similarity learning recently attracted a lot of attention from the research community. This paper provides an interesting view to study generalization under similarity learning.
- Experimental results are plentiful to justify claims.

**Weaknesses**

- The motivation is unclear. The paper begins with contrastive learning. However, it mainly targets simplified Siamese-type networks that are obviously different from recent popular contrastive learning.
- Both writing and organization need to be improved. Reviewers and AC find that it is somewhat hard to follow this paper. Although in the rebuttal, the authors provide responses for clarification, there are still remaining concerns by reviewers. The current paper does not seem to be easily improved by superficial modifications.